# Multi-System Urban Waste-Energy Self-Circulation: Design of Urban Self-Circulation System Based on Emergy Analysis

**DOI:** 10.3390/ijerph18147538

**Published:** 2021-07-15

**Authors:** Xiaoyu Xu

**Affiliations:** Landscape Architecture Department, Rhode Island School of Design, Providence, RI 02903, USA; xxu0004@gmail.com; Tel.: +1-401-215-0246

**Keywords:** urban systems, sustainability, emergy evaluation, ecosystem services, energy reconfiguration, energy-waste flow cycle, regional ecosystem management

## Abstract

The current worldwide state of energy scarcity and low waste utilization has led to a decrease in the supply of ecological services, something that seriously affects the development of cities. In this study, we propose an urban self-circulation design based on multiple systems within the traditional biogas, wetland, rainwater, solar power, and urban farm systems framework to achieve effective improvements in urban waste utilization and the optimization of the urban waste–energy flow cycle. Emergy conversion is used to evaluate system optimization, and the simulation results show that the novel proposed system can effectively improve urban waste utilization with an energy output rate of 3.18 × 10, an environmental load of 4.27 × 10^−2^, and a sustainability index of 7.45 × 10^2^ in the core system; additionally, it can improve resource utilization of small-scale cities with an energy output rate of 1.85 × 10^0^, an environmental load of 1.20 × 10^0^, and a sustainability index of 1.54 × 10^0^ in the total system. The inter-system energy flow model can significantly optimize urban energy systems based on ecological models with low-emergy resource input, including biogas systems and urban farm systems. This model can reduce the environmental load and effectively compensate for the reduced supply capacity of ecosystem services caused by urbanization, making it suitable for extension to other small-scale built environments that are relatively independent and rich in natural resources.

## 1. Introduction

As urbanization continues, human demand for ecosystem services, especially energy supply, has increased dramatically.

Ecosystems provide a range of services that are essential to supporting economic performance and human well-being; these services are referred to as ecosystem services [1,2,3]. Ecological services represent the contribution of ecosystems to human well-being and are therefore defined in terms of their specific benefits to individuals or society [4]. In addition to regulation and cultural services, the most important part of ecological services is how they supply energy and materials—including food, water, and oxygen—to cities. The resources needed for human survival and development are ultimately derived from natural ecosystems.

However, the urbanization process has largely led to a decrease in the supply capacity of ecological services. In the urbanization process, forest, pastoral and agro-ecosystems are being transformed into human-dominated ecosystems at an unprecedented scale and speed. This process inevitably disrupts the structure and function of pre-existing ecosystems and leads to the loss of ecosystem services [5,6]. Likewise, rapid and large-scale urbanization resulting in dramatic land use conversion has a dramatic impact on ecosystem services [7]. Numerous studies have shown that urbanization is one of the main drivers of ecosystem service change [8,9,10]. The reasons for this can be summarized in terms of unsustainable waste–energy flows between cities and ecosystems: the expansion of urban demand for energy and land conversion to the built environment lead to a decline in energy supply for ecosystem services; waste emissions from cities to the natural environment further damage the environment and lead to a further decline in energy supply for ecosystem services. Since a sustainable energy cycle cannot be formed between cities and the ecological environment, the contradiction between energy shortages and the considerable waste produced by cities becomes more and more prominent. It is widely believed that as urbanization continues, the pressure applied to ecosystems will increase. Sustainable urban development has become one of the most pressing issues facing humanity today [11].

Some scholars suggest that the relationship between ecosystem services and urbanization is not necessarily negative, and that said relationship is complex [7,12,13]. Picchi et al. suggest that “the impact of renewable energy technologies on the landscape infrastructure and the delivering of material or immaterial benefits or ecosystem services can be critical” [14]. The use of renewable energy sources can compensate for a lack of ecological services supply; however, the optimal option is to use renewable energy sources in tandem with waste, thus mitigating the negative impacts of existing unsustainable waste–energy streams. A better conversion of waste into usable energy and the reconstruction of urban waste–energy self-circulation systems would be effective solutions to the threat posed by urbanization to the sustainable development of cities and regions.

There have been attempts to create specific kinds of self-circulation system that reconstruct waste–energy flows. One widely used example is the rice–fish coculture system, which is thought to reduce the use of chemical fertilizers in the paddy subsystem, improve land productivity, maintain soil fertility [15], and reduce fish diseases in the fish pond subsystem so as to improve yields [16]. The underlying principle is that the paddy subsystem has a plant water purification effect on the water in the fish pond subsystem, while waste products (such as manure output from the fish pond subsystem) become nutrient inputs to the paddy subsystem. Other, similar systems use animal manure as a renewable resource and leverage intra-system energy flows among multiple agricultural production subsystems, including those that feature biogas, hot peppers [17], and the combined production of rice and ducks [18].

However, the waste–energy flow of such self-circulation systems is mostly limited to small-scale agricultural production and does not extend to other types of energy systems. In fact, the same substances are often present in the input and output forms of cities’ various energy series, such as water, electricity, and organic matter. Moreover, these systems have not yet incorporated the most important energy system into themselves—namely, the built environment, the system that consumes and exports the most energy–waste. This lack of incorporation allows for more possibilities for exploration in this direction. For example, it is possible to integrate smaller-scale urban areas into various types of energy systems that comprise a self-circulation system.

In response to the modern urban state of energy scarcity and low waste utilization, we design and examine in the current study a multi-system waste–energy-based Urban Self-circulation System. Unlike traditional urban energy systems that operate separately, the proposed Urban Self-circulation System uses circulation and the balance of energy as the emitting point to build energy and material flows among internal subsystems.

The waste generated by urban dwellers on a daily basis mainly consists of organic waste and wastewater, and thus the core Urban Self-circulation System—which aims to improve waste utilization—should consist of subsystems that treat both. The biogas system (which converts organic waste into biogas and fertilizer) and the constructed wetlands (which purify wastewater) are currently more technologically mature and relatively more ecologically-oriented energy systems for treating these two types of waste [19,20]. For this reason, we use a biogas subsystem and a constructed wetland subsystem as core systems by which to handle urban waste streams. 

However, large losses will inevitably occur during the energy cycle. Therefore, in order to achieve higher energy efficiency, this system should contain energy resources such as water and electricity, both of which are essential for daily consumption by city inhabitants but also for maintaining subsystem operations. The solar power system (which converts solar energy into electricity) and the rainwater garden system (which converts rainwater into municipal water) rely mainly on renewable energy sources such as sunlight and rainwater and are considered more ecologically-oriented energy production systems [21,22]. The biogas system has been widely used in combination with agriculture [19]: the water, fertilizer, and electricity output from the four aforementioned systems can be used as renewable energy inputs in agricultural production, making an urban farm as a subsystem of the system possible. Thus, the rainwater garden subsystem, solar power subsystem, and urban farm subsystem, together with the two core subsystems, form a complete Urban Self-circulation System that helps achieve a cycle of energy balance (Figure 1). Under this system, urban waste utilization is improved and urban energy–material flow cycle processing optimized.

The current study uses an emergy analysis method based on emergy calculation to evaluate system optimization; this study ultimately proves that this system achieves higher utilization in urban energy reuse and can play a role in reducing the environmental load caused by cities. In this way the system helps achieve sustainable urban and regional development.

Especially for small-scale built environments that are relatively independent and near to the natural environment, it is of great significance to establish a self-circulating urban system by making full use of local climatic conditions and natural resources. For example, downtown Providence in Rhode Island, United States, a city of 10,000 people and relatively simple industries, is located next to the Providence River. Built environments that meet the same criteria include many suburban communities in developed countries, emerging towns in developing countries, and villages. The proposed Urban Self-circulation System is characterized by waste treatment, energy circulation, and a combination of climate environment use and making full use of waste, wastewater, solar energy, rainwater, and other resources generated by the city as energy inputs—inputs that, after treatment, will ultimately feed back into the city in the form of energy. In this way, the self-circulation of waste and resources is realized.

The current study also evaluates the urban self-circulation design proposed herein, using the emergy analysis method; to this end it uses emergy output, environmental load, and sustainability as indicators. Analytical results show that the system offers several advantages, including low total cost, low environmental pollution, and high output. This system is validated using the downtown area of Providence, Rhode Island, as a case study. Ultimately, our findings point to the value of this scheme for replication in small and medium-sized towns and cities.

## 2. Research Objectives and Methods of Urban Self-Circulation System

### 2.1. The Total Self-Circulating System

The urban recycling system studied herein considers the traditional urban model, which features a low waste utilization rate and a lack of sustainable energy supply; it also establishes an urban self-cycling design based on a multi-system waste–energy flow (Figure 1). The main objectives underpinning this system are as follows:Improvements to the waste utilization rate, to preclude the ecological damage otherwise caused by urban waste;Improvements to the resource utilization rate, to reduce waste and pollution during the resource production process.

In terms of waste utilization improvements, the proposed urban self-recycling system contains two waste treatment subsystems: a biogas system (organic waste to biogas) and a constructed wetland system (wastewater to municipal water). In particular, the biogas system reuses organic waste and treats it as biogas through anaerobic digestion; it is then leveraged as an energy source for the city (Figure 2). The constructed wetland system, on the other hand, reuses wastewater and treats it as municipal water that flows through wetlands and is used as a source of urban water (Figure 3).

In terms of waste–energy flow circulation, this urban model contains another three subsystems: a rainwater garden system (rainwater to municipal water), a solar power system (solar power to electricity), and an urban farm system (food).

Urban waste–energy recycling systems target urban waste and energy, and use waste to achieve a sustainable material–energy cycle; in this way they create more renewable energy and mitigate the environmental damage otherwise caused by urban waste outputs. Such systems include those that treat municipal organic waste and wastewater, and produce resources such as food, water, and electricity; they also play an active role in urban sanitation and air purification. The proposed system is significant, in that it fulfills these roles at a lower cost, with a lower environmental load, and with a higher emergy conversion factor than traditional means through a total system of distributed treatment.

The proposed Urban Self-circulation System offers three major advantages, as follows. 

It makes full use of local climate and environmental characteristics, which makes it especially suitable for small and medium-sized livable cities that feature abundant sunshine and rainfall. It can therefore serve as a model in the development of such cities;It is highly effective and sustainable. Renewable resources are fully utilized in this system, with solar energy, rainwater, and municipal waste being the main energy inputs; in this way, input costs are reduced while ensuring sustainable energy outputs;This system is nonindustrial and causes little damage to the environment. The system operates in a nonindustrial way to control the discharge of harmful substances from the city to the environment, thus achieving a low environmental load. The rain gardens, constructed wetlands, and urban farms within the system provide additional environmental benefits as parts of the landscape itself.

#### 2.1.1. Biogas Subsystem

Biogas subsystems can reuse municipal organic waste by passing organic waste from the city into the biogas system. There, various organic substances are degraded by microorganisms and converted into biogas and inorganic substances such as nitrogen and phosphorus.

Biogas systems are considered more ecological and sustainable energy systems, given the benefits inherent in their use of anaerobic digestion; these include the production of renewable energy and liquid manure, and the reduction of organic pollutants by 50–90% [23]. The use of methane-rich biogas also reduces the amount of greenhouse gases released into the environment, relative to conventional manure management systems [19]. Other benefits of biogas systems include reductions in the odors and pathogens associated with livestock manure [24,25]. Existing emergy analysis studies show that biogas systems with anaerobic digestion systems rely on renewable energy sources for more than half of their inputs [19].

A biogas subsystem offers significant advantages in an energy cycle, as follows. 

High conversion rate. Biogas systems can effectively convert organic waste into usable materials, including biogas and inorganic nitrogen and phosphorus;Clean energy source. Methane-rich biogas is a relatively clean energy source, and it can be used in combustion to generate heat or power. It is less damaging to the environment than fossil fuels and can effectively reduce organic pollutant emissions;Smaller emissions. Biogas systems have the advantage of emitting less greenhouse gas and transmitting fewer harmful pathogens than traditional manure systems (e.g., the transmission of pathogenic bacteria in manure is blocked).

#### 2.1.2. Constructed Wetland Subsystem

Constructed wetland subsystems can be used in the provision of clean water to a city. After wastewater passes from the city into the constructed wetland subsystem, it is treated by microbial, biological, physical, and chemical means, whereupon it is converted into water that can then be transformed into useable urban water.

Studies have concluded that constructed wetland systems have the advantages of high treatment capacity, high treatment efficiency, and low cost, compared to traditional wastewater treatment systems. Compared to wastewater treatment systems using a cyclic activated sludge system (CASS) and the conventional activated sludge (AS) process, the capital cost per cubic meter of wastewater treated per day in constructed wetlands is reduced by half [26,27]. Due to the high consumption of CASS/AS treatment in terms of purchased resources (e.g., fuel, electricity, steel, and cement), the cost difference between constructed wetlands and conventional treatment is even larger when one considers total life-cycle consumption.

In addition, the energy used to operate and maintain constructed wetlands constitutes only a small percentage of the total energy input (7.4%), with less electricity consumption translating into lower fuel consumption and greenhouse gas emissions. An emergy analysis study of wastewater treatment systems shows that constructed wetlands are less dependent on external resources and imported energy than CASS and AS systems [20].

Finally, while conventional wastewater treatment systems generally tend to be built near metropolitan areas, constructed wetlands can be built in rural areas, for example [28]. This factor aligns more with the geographical conditions this study proposes.

In summary, compared to traditional wastewater treatment systems, constructed wetland systems have the advantages of high treatment capacity, low cost, and low environmental load.

#### 2.1.3. Rainwater Garden Subsystem

The main role of the biogas and constructed wetlands subsystems in the Urban Self-circulation System is waste treatment. However, during the waste treatment process, the available energy output is inevitably less than the energy initially put into the city. For example, the constructed wetland treatment can reclaim 83% of the water available at input. Therefore, we need more subsystems by which to put additional energy into the city and thus maintain a balance between inputs and secondary inputs.

A rainwater garden is a rainwater-harvesting system wherein the obtained water can compensate for losses incurred during wastewater treatment. This system maintains the advantages of being pollution-free and low cost while capturing water resources efficiently. Rainwater garden systems have been actively adopted by many institutions and municipalities, on account of their ability to enhance the retention, infiltration, and reuse of stormwater in the urban landscape. In addition to stormwater management, rainwater gardens can provide various benefits, including mitigation of the urban heat island effect, energy use reductions, improvements in air and water quality, carbon sequestration, benefits to human physical and mental health, access to recreational opportunities, and improved biota habitat [21]. Many of these additional benefits can play a role in mitigating climate-change impacts in urban environments (or help us adapt to them) [29] and positively impact local property values [30].

#### 2.1.4. Solar Power Subsystem

Solar power systems are waste-free, fuel-free energy-generation systems that have significant advantages over traditional thermal power systems that not only require the burning of fossil fuels but also emit harmful gases. Solar power systems are considered more environmentally friendly than such conventional power generation systems. Photovoltaic (PV) system operation is driven mainly by renewable energy (i.e., sunlight), while thermal power generation requires fuel oil (a nonrenewable energy source) as a feedstock [31]. Moreover, during the operational phase, thermal power generation requires energy and service inputs, plus renewable energy (in the forms of cooling water and wind energy to diffuse pollutants); solar power systems, on the other hand, make no use of these energy resources [22].

Stable operation of the urban power grid is a necessary response to urban development, and existing studies sufficiently demonstrate that solar power is an effective and economical supplement to electricity. The following sections evaluate the importance of solar power systems in maintaining the stability of the overall system.

#### 2.1.5. Urban Farm Subsystem

A major factor that exacerbates the environmental impact of cities is the transport of food supplies to urban areas [32], as urban populations are often distant from food production traditions and their environmental impact [33,34]. Therefore, urban agriculture may help reduce the impact of cities and increase global sustainability. Many studies conclude that urban agriculture not only reduces food miles, but also contributes to the social, economic, and environmental sustainability of cities by enhancing biodiversity, protecting urban soils, improving the microclimate, indirectly improving water management, providing access to nutrients and waste recycling, reducing atmospheric pollution and global warming impacts (e.g., reduced transport of food, increased carbon dioxide uptake), and potentially enhancing environmental awareness [35,36].

In contrast to traditional food production systems, urban farm systems are not highly dependent on imported and nonrenewable resources (e.g., transportation fuels, machinery fuels, pesticides, and fertilizers) or on additional services needed to transform raw materials into various products [37]. Instead, urban farms can omit such energy inputs by using biogas or local composting to generate fertilizers, organic methods to reduce pesticide use, and the like.

An urban farm is a system created to replenish urban food both locally and in surrounding areas. For a large self-cycling urban system, the establishment of an urban farm offers the following advantages:Cost-effectiveness. More food is sourced locally, which reduces the energy waste and waste generation associated with food transport;Environmental optimization. Agricultural farming within the urban space is conducive to improving the urban environment;High degree of recyclability. Urban farming is a food system that helps advance the urban waste–resource–energy cycle—especially when combined with a broader urban waste–energy recycling system where the electricity and water needed for urban farming can be imported at no cost.

The following sections evaluate the importance of including urban farms in maintaining the stability of the overall larger system.

### 2.2. Transfer and Feasibility of Energy Self-Circulation

Figure 1 illustrates the main energy flows among the aforementioned subsystems. In the operational phase, external inputs include urban waste and wastewater as well as renewable energy sources such as solar energy and rainwater. The system outputs consist of urban water, natural gas, electricity, food, and other urban resources that can be used directly. These conditions result in an energy–matter flow cycle within the city.

Self-circulation is considered feasible when the output energy–matter flow is equal to the initial energy–matter flow.

The proposed Urban Self-circulation System for urban energy flow optimization includes energy flows among subsystems, together with the addition or replacement of more advanced subsystems. In the proposed model, the water generated by the constructed wetland system can be used in other systems (such as the urban farms and rainwater gardens); meanwhile, the electricity generated by the solar power system and the nitrogen and phosphorus inorganic matter generated by the biogas system can also be used in the urban farm system. Ultimately, the five systems work together to export various resources that the city needs. In other words, by working together, these subsystems can take resources that would otherwise go to waste and renew and reuse them, thus creating a feasible urban self-recycling system.

### 2.3. Advantages of a Self-Cycling City Model

The urban resource self-recycling design described herein more fully achieves the realization of urban self-recycling than any system previously proposed. It does so by creating a more complete model of a superior waste–energy cycle by which to address the existing urban concerns of waste underutilization and the lack of sustainable energy supplies. The proposed system offers the following major advantages.

Scalability. The system makes full use of local climate and environmental characteristics. It is suitable for small and medium-sized livable cities with abundant sunlight and rainfall, and can be used as a model for developing such towns.Low cost of sustainable operation. Waste and wastewater in cities are treated by discrete subsystems so that resources can be reused, effectively reducing costs. It also produces additional resources, such as food and energy.Environmentally friendly. Increased waste utilization reduces environmental damage that would otherwise occur, and enhanced access to solar energy supplies reduces emissions of polluting gases.

## 3. Evaluation and Validation

### 3.1. Research Methodology: Evaluation of Urban Self-Cycle Systems Based on Emergy Analysis Methodology

Emergy assessment theory is based on the assumption that the value of a resource is proportional to the energy required to produce it [38]. In other words, it represents an effort to evaluate the real wealth contributions of natural environments, and it uses energy as a common currency to compare vastly different resources [39]. Given the energy balance between urban and natural environments—which is the main concern of this study—and the many different types of energy forms involved in the proposed Urban Self-circulation System, the use of an emergy assessment is most reasonable. An emergy assessment can quantify the input, consumption, and output of energy and materials in a system, and thus determine that system’s efficiency, sustainability, and degree of environmental impact.

The basic principle of energy value assessment is that one energy source is not qualitatively equivalent to another when one takes into account indirect energy consumption in the production process. To compare different energy sources and materials in the production process, we convert all system inputs into the same unit of energy: the solar joule (sej). In emergy assessment theory, solar energy is defined as “the available solar energy used up directly and indirectly to make a service or product.” The units of solar emergy are solar emjoules (sej), which are defined as one joule of solar radiation received by a substance. This definition considers the fact that the ultimate source of most energy and materials on Earth is the sun.

In the emergy concept, the quality of energy is expressed in terms of transformity, which is defined as “the quotient of a product’s emergy divided by its energy” [38]. In a series of related studies, scholars have evaluated various systems by determining the transformity of various energy forms, including rainwater, sewage, organic waste, and electricity.

The emergy analysis process is divided into three steps. In the first step, we construct a system diagram to organize our knowledge concerning the system’s major components and processes.

In the second step, we construct emergy evaluation tables. In emergy calculation, the energy flow invested in the system is divided into three types: renewable local resources (R), nonrenewable local resources (N), and purchased resources (F). Here, “local” denotes a resource acquired within the system. For example, while sunlight and rainwater are common renewable resources, resources that are obtained locally but whose regeneration speed cannot categorize them as renewable in system operations are considered nonrenewable resources. One common example is soil loss. Resource investments from outside the system are called purchased resources, or imported resources, and in the system they too are considered nonrenewable.

In the current study, the traditional single energy system is compared to the proposed Urban Self-circulation System approach, the latter of which can change a portion of the nonrenewable local resources (N) and purchased resources (F) into renewable local resources (R) in the emergy evaluation tables, thus affecting the evaluation results.

The raw data in each row were obtained from statistical references and published literature on the systems in question, for which material balance calculations had been performed. The transformity values of various projects were obtained from previous energy assessment studies [38]. The emergy of each item was then obtained by multiplying the raw data by its conversion rate. 

The third step is to calculate several indices in a contingency evaluation table. There are many indices by which to evaluate the emergy index of a system. Taking into account the urban energy sustainability and environmental impact studied herein, this study uses three emergy indexes—namely, the emergy output rate (EYR), environmental load rate (ELR), and emergy sustainability index (ESI)—each of which we describe below.

The emergy output rate is the ratio of the total emergy input (U) of the system to the purchased resources, namely EYR = U/F. The emergy output rate is defined as the output–cost ratio [40], and it is used to measure economic benefits. The emergy output rate always exceeds 1, and the larger it is, the larger the profits are [41].

The environmental load rate is the ratio of the input of nonrenewable resources and purchased resources to renewable resources, namely ELR = (N + F)/R. It represents the degree to which a system relies on nonrenewable resources, which in turn represents its pressure on the environment. Therefore, the smaller the environmental load rate is, the less the system will damage the environment [41].

The emergy sustainability index is the ratio of emergy output rate to the environmental load rate, namely ESI = EYR/ELR. It represents the economic output of the system per unit of environmental load. When the sustainability index is less than 1, the system is a net consumption process. The higher the sustainability index, the more sustainable a system is [41].

As the values of nonrenewable local resources (N), purchased resources (F), and renewable local resources (R) change, the final emergy output rate (EYR), environmental load rate (ELR), and sustainability index (ESI) will also change. This means that the proposed Urban Self-circulation System affects the efficiency of the urban energy output, the pressure placed on the environment, and system sustainability.

### 3.2. Assessment Process

The assessment model proposed herein consists of three aspects: an assessment of each discrete subsystem, an assessment of the core Urban Self-circulation System containing these two discrete subsystems, and an assessment of the proposed self-circulating urban system containing multiple systems.

The discrete subsystems assessment examines the output efficiency, environmental load, and sustainability of the five subsystems selected for the system itself, and to establish a more rigorous evaluation for each discrete subsystem. The core Urban Self-circulation System assessment, on the other hand, focuses on whether the disposal of urban waste is economically and environmentally feasible. Finally, the evaluation of the complete self-recycling city system primarily considers the feasibility of a self-recycling city, and comprehensively considers several indicators to demonstrate the value of the model proposed herein.

### 3.3. Data Analysis and Pre-Processing Based on Downtown Providence

The current study references the downtown area of Providence, Rhode Island, and its organic waste, sewage, electricity consumption, water consumption, and food consumption [42]. The data are quantified and used in the model analysis. When one considers the proposed Urban Self-circulation System overall, the amount of waste treatment and energy production in each subsystem needs to follow a certain ratio. Table 1 presents the results of this analysis.

The downtown area of Providence is a standard and complete area, and is representative of many small to medium-sized towns: there, 10,000 people live, spend money, and work. The downtown area has a robust mix of commercial, office, residential, and educational functions where people can live a regular city life.

The city is located in a livable environment with plenty of sunlight and rainwater, but the city’s energy supply mainly comes from outside. In other words, Providence has an energy-consuming, waste-generating system, which makes it suitable for validating the proposed model.

Providence’s energy consumption and waste generation are dominated by the population that lives within it and the urban facilities that support it, rather than large industrial and commercial facilities. This setting aligns with the small and medium-sized towns for which the proposed Urban Self-circulation System is intended.

If the proposed Urban Self-circulation System proves to be effective in addressing waste reuse in this area—and even in addressing the city’s energy supply—it means that we can make the energy–material cycle of this and similar cities more sustainable, and thereby preclude the environmental destruction typically caused by such cities.

### 3.4. Emergy Calculation

#### 3.4.1. Before and after the Operation of Each Subsystem

We use biogas production from small-scale agricultural digesters as the object of study for the organic waste treatment system. In an anaerobic digester, energy in the form of methane-enriched biogas is produced through the microbial degradation of various types of organic matter inputs, but most commonly livestock manure. During anaerobic digestion, complex organic molecules such as carbohydrates, proteins, and fats are transformed through a multi-step microbial-mediated biochemical pathway. The end products of this process include methane, carbon dioxide gas, and inorganic forms of nitrogen and phosphorous [43].

In addition to the manure, water, and sunlight used to generate the products, energy resources put into the system include steel, zinc, and the like used to construct the equipment, and electricity used to operate the equipment. These resources are incorporated into the energy calculations based on an average equipment life of 20 years (Figure 4).

Case data are drawn from a 2011 emergy evaluation study of Taiwanese model plug-flow anaerobic digesters located on la Región Tropical Húmeda University campus in Guácimo, Limón, Costa Rica, within a small ranch of Taiwanese model plug-flow anaerobic digesters. The campus is located at an altitude of 50 m above sea level with an average annual temperature range of 25–30 °C. The digester consists of polyethylene tubular bags with a thickness of 0.2 mm, a total length of 63 m, and a total volume of 146 m^3^; their average residence time is 26.5 days [19].

Downtown Providence generates an average of 5.74 × 10^9^ g of organic waste per year [42]. This figure is multiplied by the transformity of 9.70 × 10^7^ sej/g given by Bastianoni and Marchettini in 2000 to obtain an additional 5.57 × 10^17^ sej of emergy per year. Together with the 4.68 × 10^10^ J of electricity from the solar power subsystem (equivalent to 7.72 × 10^15^ sej), a total of 5.65 × 10^17^ sej of purchased resources are converted each year in the system into renewable resources, thus influencing the final assessment results. The system produces 1.23 × 10^13^ J of biogas, as well as nitrogen and phosphorus that could be used as fertilizer.

With the biogas system operating alone, the energy output rate would be 1.05 × 10^0^, the environmental load rate 1.82 × 10, and the sustainability index 5.80 × 10^−2^. However, if the biogas system were to be integrated into the Urban Self-circulation System, the energy output rate would become 2.27 × 10 and the environmental load rate 4.61 × 10^−2^ as the organic matter that was originally wasted and the electricity resources that needed to be purchased become part of the recycling stream. The sustainability index would become 4.92 × 10^2^ (Table 2 and Table 3).

The current study uses constructed wetland systems as a water depuration system. Constructed wetlands are artificial wastewater treatment systems consisting of shallow ponds or channels into which aquatic plants have been planted; these systems leverage natural microbial, biological, physical, and chemical processes to treat wastewater [49]. Downtown Providence is adjacent to the Providence River and has a large amount of vacant land along the surrounding shoreline that is suitable for constructed wetlands.

In addition to the wastewater, rainwater, and sunlight used to generate the products, the energy resources put into the system include soil, vegetation, fillers, gravel, and the like used to construct equipment, and electricity and compost used to operate the equipment; these are incorporated into the energy calculations (Figure 5).

Case data are drawn from a 2009 emergy assessment study of a constructed wetland on the Longdao River in Beijing. The daily runoff volume of the Longdao River in Beijing is 16,000 m^3^. The wetland system at the estuary was constructed in late 1995 and consists of four parallel gravel units, each with a length of 50–25 m and a horizontal subsurface flow of 0.6 m. The Longdao River constructed wetlands located at the estuary contain a 600 m^2^ vertical subsurface flow vegetated bed with a daily capacity of 200 m^3^ of wastewater. It is estimated that the wastewater treatment plant will have a service life of no fewer than 20 years [20].

Downtown Providence generates an average of 2.97 × 10^16^ J of wastewater per year [42]. This figure is multiplied by the transformity 3.80 × 10^6^ sej/J given by Björklund et al. in 2001 to derive 1.13 × 10^23^ sej of emergy per year. Although the emergy evaluation tables in both cases are identical, when operating alone or as a subsystem of the proposed Urban Self-circulation System, the output product water (2.48 × 10^16^ J/year) can be put into other subsystems as a renewable resource. In this case, the emergy output rate is 3.19 × 10, the environmental load rate is 4.27 × 10^−2^, and the sustainability index is 7.47 × 10^2^ (Table 4).

The current study focuses on rainwater gardens as a research object in examining supplementary energy (water) systems. Rainwater gardens are artificially excavated shallow depressions that are used to collect and absorb rainwater from the roof or ground, purify it with a combination of plants, sand, and soil, and discharge it to municipal water sources.

In addition to the municipal water, rainwater, and sunlight used to generate the products, the energy resources put into the system include soil, vegetation, various equipment construction materials, and the machinery and tools used to operate the equipment; these are all incorporated into the energy calculation (Figure 6).

For our emergy evaluation study, we make use of case data from a 2011 study of over 100 green infrastructure projects built between 2010 and 2013 in Syracuse, New York, United States, as part of the Save the Rain program. The 45 rainwater gardens included in this study provided the data for the study [21].

Fertilizer, a product of the biogas subsystem, and municipal water, a product of the constructed wetland subsystem, constitute the main differences between the two data sets: 3.25 × 10^9^ g of fertilizer was multiplied by the 1996 transformity given by Odum of 8.28 × 10^9^ sej/g to obtain 2.69 × 10^19^ sej of emergy per year; 9.22 × 10^10^ J of municipal water was multiplied by the transformity 4.31 × 10^5^ sej/J given by Brown and Buranakarn in 2003 gives 3.97 × 10^16^ sej of emergy per year. In total, the system converts 2.69 × 10^19^ sej of purchased resources into renewable resources each year, thus affecting the final assessment results.

We learned from the constructed wetland system that the rainwater garden system needs to reproduce 4.88 × 10^15^ J of water. To produce these quantities of municipal water for cities, the emergy output rate of the rainwater garden operating alone would need to be 1, while the environmental load rate 7.63 × 10^2^, and the sustainability index 1.31 × 10^−3^. However, when a rainwater garden system is incorporated into the proposed Urban Self-circulation System and we consider the municipal water (originally purchased as a resource) to instead be a renewable resource, the emergy output rate is 1, the environmental load rate is 7.69 × 10^2^, and the sustainability index is 1.31 × 10^−3^ (Table 5 and Table 6).

We selected a small-scale urban farm as the object of study for a supplementary energy (food) system. Such farms use less mechanical inputs and are considered more environmentally friendly [35].

In addition to municipal water, seeds, rainwater, and sunlight are used to generate the farm products. The energy resources put into the system include soil, vegetation, equipment construction materials (e.g., wood and the like), compost, pesticides, and electricity used to operate equipment, all of which are incorporated into the energy calculation (Figure 7).

The case data comes from a 2017 emergy evaluation study of a local urban farm in Detroit that, at 1712 m^2^, is relatively small. Most of the compost comes from local Detroit farms and gardens. Using organic methods as the basis for food production, the farmers refrain from using any synthetic fertilizers and use only the organic pesticide neem oil on the transplants before planting. To supplement rainwater, two water-harvesting systems with two tanks each are used to conserve water throughout the growing season [35]. After serving as a subsystem of the proposed Urban Self-circulation System, its compost, water, and electricity will derive from the output products of a biogas subsystem, constructed wetland subsystem and rainwater garden subsystem, and solar power subsystem, respectively.

Fertilizer, municipal water and electricity constitute the main differences between the two data sets: 3.35 × 10^10^ g of fertilizer multiplied by the 2011 transformity 2.42 × 10^8^ sej/g given by Zhang et al. yields 8.11 × 10^18^ sej of emergy per year; 6.05 × 10^10^ J of municipal water multiplied by the transformity 4.31 × 10^5^ sej/J given by Brown and Buranakarn in 2003 gives 2.61 × 10^16^ sej of emergy per year; and 3.37 × 10^12^ J of electricity multiplied by the transformity 1.66 × 10^5^ given by Odum in 1996 gives 5.59 × 10^17^ sej/J of emergy per year. In total, the system converts 8.70 × 10^18^ sej of purchased resources into renewable resources each year, thus affecting the final assessment results.

The annual food demand in Downtown Providence is 9.54 × 10^6^ kg. If the urban farm system were used to fulfill the supply, the emergy output rate would be given while considering the purchase of municipal water, electricity, fertilizer, and other resources; this value is 1.03 × 10^0^, while the environmental load rate is 3.58 × 10, and the sustainability index is 2.87 × 10^−2^. When the urban farm system is incorporated into the Urban Self-circulation System, the emergy output rate becomes 1.35 × 10, the environmental load rate 8.01 × 10^−2^, and the sustainability index 1.68 × 10^2^ (Table 7 and Table 8).

In the current study, cadmium telluride (CdTe) thin-film PV electricity production was chosen as the object of study with regard to supplementary energy (electricity) systems. Among all PV technologies currently available, CdTe thin-film PV carries the smallest carbon footprint, uses the least amount of water, and boasts the shortest energy payback period [63]. This technology is based on the use of CdTe in a thin semiconductor layer designed to absorb sunlight and convert it into electricity.

In addition to rainwater and sunlight used to generate the products, the energy resources put into the system include glass, water, copper, PV materials, and the like, which are used to construct the equipment. All of these are incorporated into the energy calculation (Figure 8).

The case data are drawn from a 2012 study evaluating PV plants per square meter of unit area [22]. The water from the constructed wetland subsystem and the rainwater garden subsystem constitute the main differences between the two data sets: 3.82 × 10^9^ g of water multiplied by the transformity 1.59 × 10^6^ sej/g given by Buenfil in 2001 derives 6.07 × 10^15^ sej of emergy per year. Each year, 6.07 × 10^15^ sej of purchased resources are converted to renewable resources, thus affecting the final assessment results.

The annual electricity demand in Downtown Providence is 2.75 × 10^14^ J, and the urban farm subsystem needs 2.79 × 10^14^ J. If a solar power generation system were used, the emergy output rate would be 1.01 × 10^0^, the environmental load rate 6.14 × 10^3^, and the sustainability index 1.63 × 10^−4^. When the solar power generation system is incorporated into the Urban Self-circulation System, the water that was originally considered a purchased resource is now considered a renewable resource, the emergy output rate is 1, the environmental load rate is 2.12 × 10^3^, and the sustainability index 4.71 × 10^−4^ (Table 9 and Table 10).

#### 3.4.2. Calculations for the Core Urban Self-Circulation System

The core urban circulation system consists of a biogas system and a constructed wetland system. The two values were summed for emergy calculation, and the emergy output rate was 3.18 × 10, the environmental load rate 4.27 × 10^−2^, and the sustainability index 7.45 × 10^2^. Compared to the results for each group calculated for the other subsystems individually and after including it in the Urban Self-circulation System, this data set represents the most economically profitable, least environmentally stressful, and most sustainable option (Table 11).

#### 3.4.3. The Urban Self-Circulation System

The Urban Self-circulation System consists of five subsystems. Following emergy calculation, we found the energy output rate to be 1.85 × 10^0^, the environmental load rate 1.20 × 10^0^, and the sustainability index 1.54 × 10^0^ (Table 11).

Compared with the indicators of each system in single operation, the Urban Self-circulation System outperforms the biogas system, the solar power system, the rain garden system and the urban farm system in all three indicators, and is inferior only to the constructed wetland system.

Our results indicate show that the model proposed in here is valuable overall. While maintaining a certain degree of environmental friendliness and economic efficiency, the system maintains a long period of self-cycling operation.

### 3.5. Overall Assessment Results of Subsystems

By analyzing the energy output rate of the five subsystems in stand-alone operations and as subsystems of the proposed Urban Self-circulation System, we found there to be a significant shift in the values pertaining to the biogas system and the urban farm system (Figure 9)—namely, the growth rates reached 2061.9% and 1210.68%, respectively. These results indicate that the proposed Urban Self-circulation System can help to increase the economic returns of these two energy systems.

Analysis of the environmental load rate revealed more significant changes in the values of the biogas system, the urban farm system, and the solar power system (Figure 10)—namely, the reduction rates reached 99.75%, 99.78%, and 65.47%, respectively. These results indicate that the proposed Urban Self-circulation System helps reduce the load of urban energy systems on the environment.

By analyzing the sustainability index, we found that the values of the biogas system and urban farm system changed significantly; those of the values of the solar power system also increased significantly (Figure 11). The values of both the biogas and urban farm systems changed from less than 1 to more than 1, which means that both systems changed from unsustainable to sustainable. The value of the solar power system was not more than 1, but increased by 188.96% compared to the original value. These results show that the proposed Urban Self-circulation System can contribute meaningfully to the sustainability of urban energy systems.

## 4. Discussion

In terms of the calculated indicators, the core urban sustainable system is one consisting of a biogas system and a constructed wetland system. This finding underscores the important role of waste treatment in the urban energy system. The adoption of more sustainable means of treating waste will effectively mitigate the ecological damage otherwise imposed by urbanization processes.

Using these indicators as guidance, one can say that the proposed Urban Self-circulation System is not optimal relative to the core urban sustainable system. In other words, the model proposed herein does not perform as well as that used to improve resource efficiency or improve waste efficiency. However, the proposed Urban Self-circulation System still has advantages over the conventional subsystem model in terms of improving resource utilization and current circumstances.

The constructed wetland subsystem has a greater impact on the total system than the other subsystems, because it treats more energy (water) than the other systems. However, a comparison of the indicators of each subsystem before and after optimization also shows the advantages of the proposed model in improving economic efficiency, reducing pressure on the environment, and improving system sustainability.

Among the five subsystems, the biogas subsystem and the urban farm subsystem showed the most significant indicator changes, followed by the solar power subsystem. The rain garden subsystem and the constructed wetland subsystem showed insignificant changes, partially because their input energy is mainly rainwater—which is a renewable resource—and so differences between before and after system implementation are not significant. Therefore, if the proposed Urban Self-circulation System model is to offer its advantages, energy flow amongst subsystems is the most critical factor.

Another reason for the difference in subsystem metrics is that the significantly altered systems have a higher proportion of input energy (in the forms of water, fertilizer, and organic matter) that can be produced in more ecological ways, and a lower proportion of energy (in the forms of industrial materials and metals) that depends on industrial production. The poor overall performance of solar power systems also stems from the fact that among the purchased resources, photoactive materials, glass, copper, and other resources contain too much emergy, and so changes in renewable resources are not on their own enough to affect the overall results. Therefore, in order to realize the advantages of the proposed Urban Self-circulation System model, the subsystem itself should adopt more ecological ways of operating and use low-emergy materials.

The emergy method offers many advantages, including metrics calibration. However, for specific systems, this algorithm may need to be adjusted somewhat to address negative impacts, such as those described earlier for stormwater systems. The exclusion of stormwater systems can bring about some improvement in metrics, given that stormwater systems may not be cost-effective. One must bear in mind, however, that the roles of stormwater systems in acting as urban landscapes, absorbing carbon dioxide, and releasing oxygen cannot be evaluated in terms of energy values; only the ability to collect water is evaluated.

At the same time, there are other more valuable aspects of urban recycling systems, such as land resources. Investments made by cities in land resources on which to build a series of waste treatment and energy yield systems are considerable, but this fact does not reflect in the energy value calculation. It should be optimized in subsequent studies to fit various models.

However, among urban recycling systems, biogas systems where the main components are buried underground, constructed wetland systems that can be used in combination with natural water bodies, rain gardens that form part of the landscape, and solar power systems that can be built on roofs are all forms of optimized land-resource use, especially in comparison to traditional power plants and wastewater treatment plants.

There is no denying that industrialization allows for the mass production of energy. The proposed Urban Self-circulation System model is ideal for small and medium-sized cities and towns, as an alternative to sustainable urban development.

## 5. Conclusions

The current study proposes a self-circulating, multi-system city design; it also evaluated this design via the emergy analysis method, which accurately reflects the actual situation.

The results show that the energy output rate of the core urban circulation system is 3.18 × 10, the environmental load rate 4.27 × 10^−2^, and the sustainability index 7.45 × 10^2^. These results are better than all the indicators of the subsystems when they operate separately, thus suggesting effective improvements in urban waste utilization. The proposed Urban Self-circulation System has an energy output rate of 1.85 × 10^0^, an environmental load of 1.20 × 10^0^, and a sustainability index of 1.54 × 10^0^; these numbers are all superior to those of all the subsystems, save for the constructed wetland system. These findings point to improvements in resource utilization, to a certain extent.

Use of the inter-system energy flow model of the proposed Urban Self-circulation System would improve the energy emergy output rate of the biogas system (2061.9%) and the urban farm system (1210.68%), thus improving economic returns. Similarly, it would reduce the environmental load rate of the biogas system (99.75%), the urban farm system (99.78%), and the solar power system (65.47%), thus reducing environmental load, and it increase the sustainability index of the biogas system (848,176%), the urban farm system (585,266%), and the solar power system (188.96%), thus improving the sustainability of urban energy systems.

Our research results show that the model offers the following advantages.

High emergy conversion. According to the results of the emergy analysis evaluation, the emergy conversion rate of this model is high, and the overall rate is better than that of the traditional model;Low cost. Circulation within multiple subsystems (waste–raw materials–energy) reduces resource waste and saves costs;Environmentally friendly. Recycling waste and utilizing clean energy reduces environmental pollution.Good system stability. A self-circulating city based on multiple systems can realize sustainable whole-system operations by leveraging internal circulation among distinct systems.

For these reasons, it is suitable to extend this model to other small-scale built environments that are relatively independent and rich in natural resources.

In developed countries, many relatively independent suburban communities formed as a result of the spread of suburbanization, while in developing countries many small towns have not yet reached scale. Additionally, each country has a larger number of rural areas isolated from their natural environments, compared to large cities. For these built environments, relying solely on large urban energy systems to support energy consumption and waste disposal may not be the best option, given the large amount of energy wasted during transmission (among other disadvantages). Moreover, some rural areas have no opportunity to share in urban energy systems.

The proposed Urban Self-circulation System model offers another option for urban sustainable development: it can serve as an energy treatment system in small-scale built environments, or as a complement to large urban energy systems. It such ways it can compensate to varying degrees for urbanization-imposed reductions in the supply capacity of ecosystem services.

## Figures and Tables

**Figure 1 ijerph-18-07538-f001:**
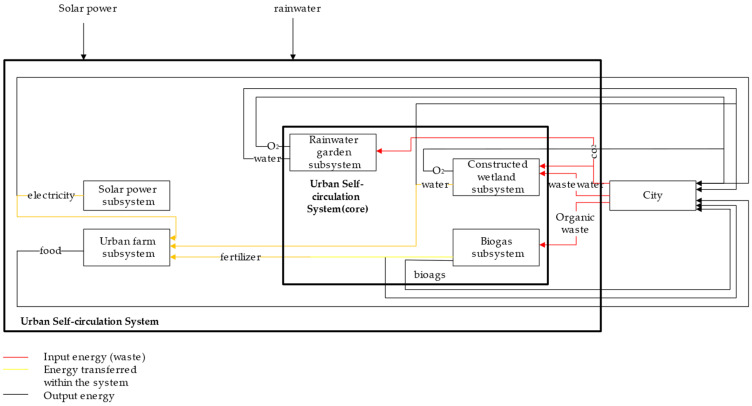
Urban Self-circulation System.

**Figure 2 ijerph-18-07538-f002:**
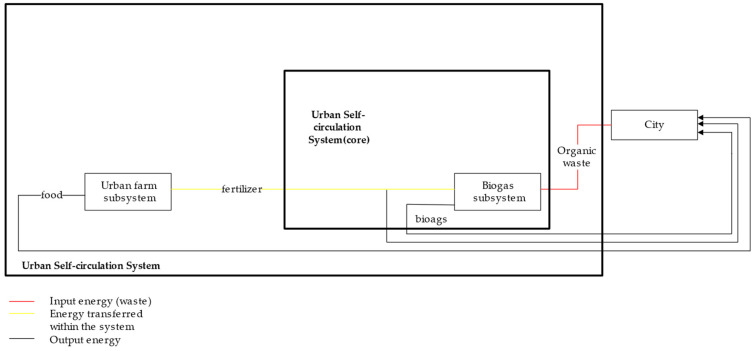
Flows of Organic Waste and Derivatives.

**Figure 3 ijerph-18-07538-f003:**
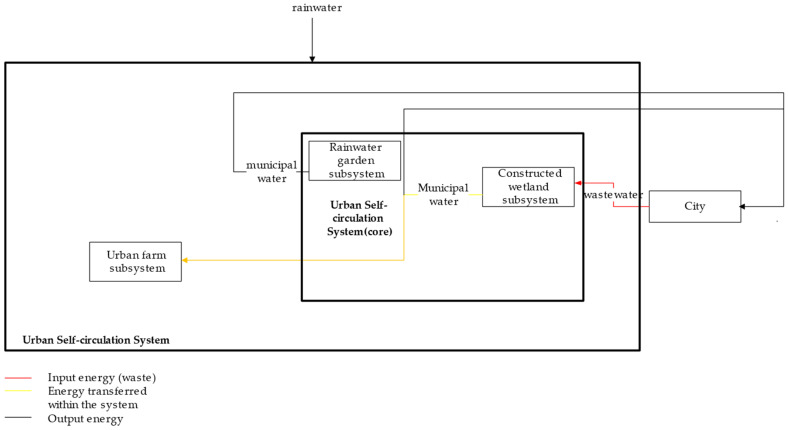
Flows of Water and Derivatives.

**Figure 4 ijerph-18-07538-f004:**
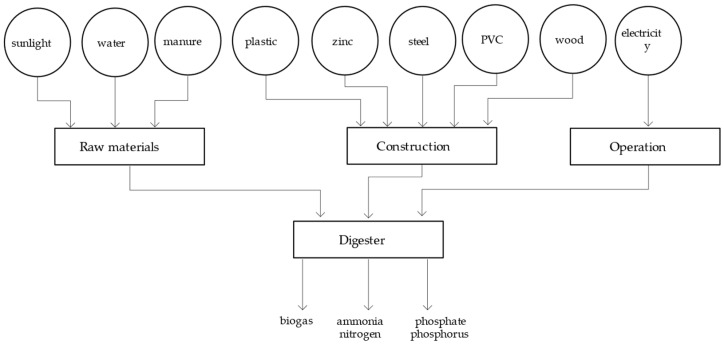
System diagram of the biogas system.

**Figure 5 ijerph-18-07538-f005:**
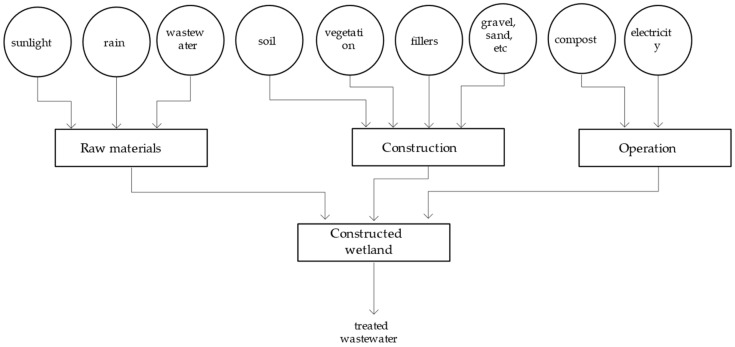
System diagram of the constructed wetland system.

**Figure 6 ijerph-18-07538-f006:**
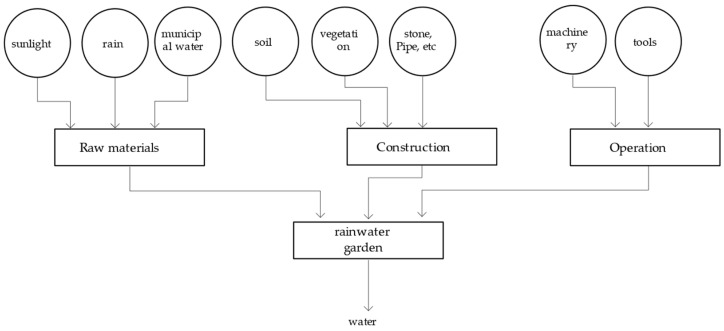
System diagram of the rainwater garden system.

**Figure 7 ijerph-18-07538-f007:**
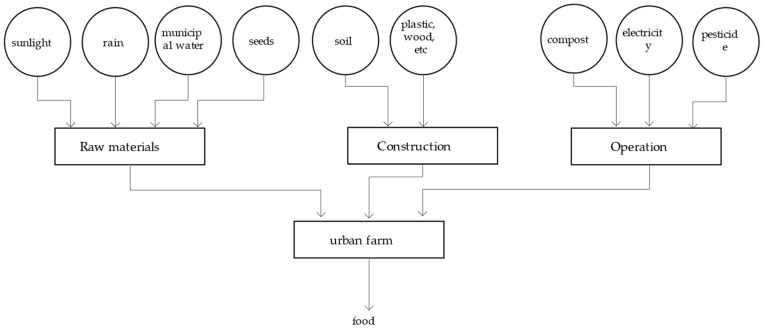
System diagram of the urban farm system.

**Figure 8 ijerph-18-07538-f008:**
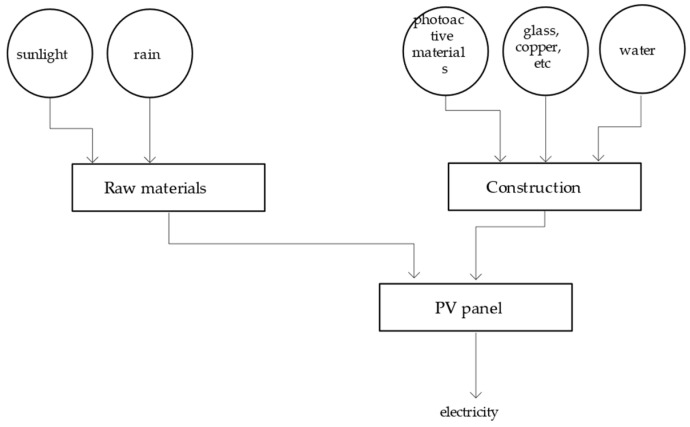
System diagram of the solar power system.

**Figure 9 ijerph-18-07538-f009:**
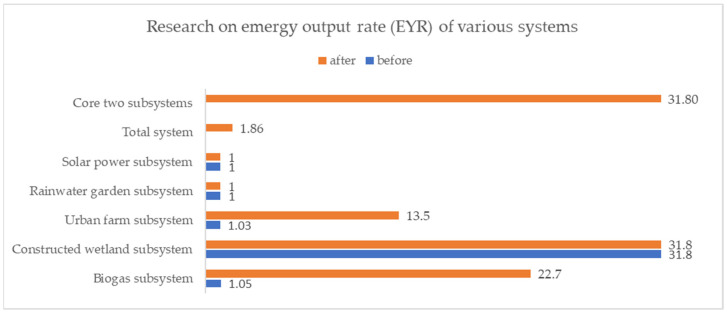
Research on emergy output rate (EYR) of various systems.

**Figure 10 ijerph-18-07538-f010:**
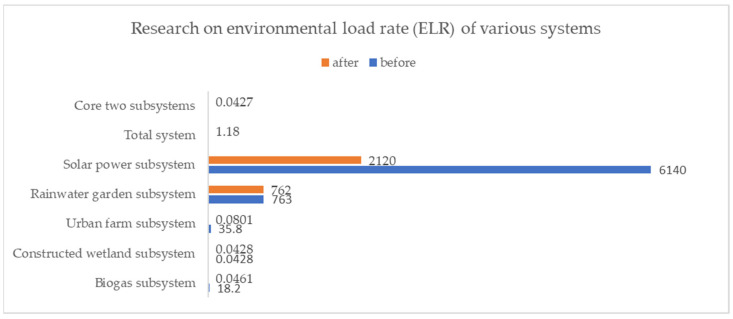
Research on environmental load rate (ELR) of various systems.

**Figure 11 ijerph-18-07538-f011:**
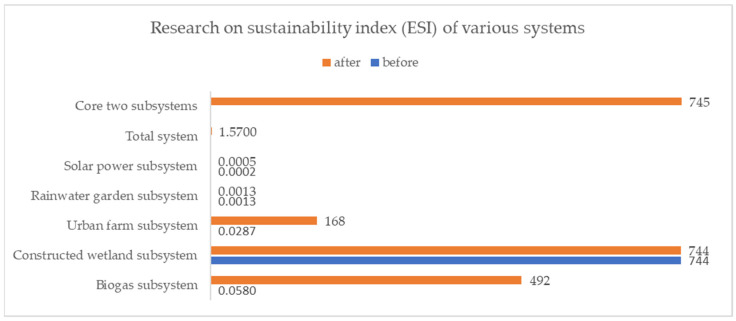
Research on sustainability index (ESI) of various systems.

**Table 1 ijerph-18-07538-t001:** Waste discharge and energy consumption in Downtown Providence.

Resource	Unit	Sum
Electricity	J	2.75 × 10^14^
Organic waste	g	5.74 × 10^9^
Water	J	2.97 × 10^16^
Food	Kg	9.54 × 10^6^

**Table 2 ijerph-18-07538-t002:** Emergy calculation of biogas subsystem (before).

	Unit	Units/Year	Transformity (sej/Unit)	Emergy (sej/Year)
Renewable resources (R)				
Sunlight	J	2.37 × 10^13^	1	2.37 × 10^13^
Water [44]	m^3^	1.01 × 10^5^	3.23 × 10^11^	3.26 × 10^16^
Electricity (from the system) [38]	J	0	1.65 × 10^5^	0
Manure (from the system) [45]	g	0	9.70 × 10^7^	0
			R	3.25 × 10^16^
Purchased resources (F)				
Electricity(purchased) [38]	J	4.69 × 10^10^	1.65 × 10^5^	7.73 × 10^15^
Plastic [46]	g	4.15 × 10^5^	5.87 × 10^9^	2.44 × 10^15^
Zinc [47]	g	2.14 × 10^5^	1.58 × 10^10^	3.38 × 10^15^
Steel [46]	g	3.58 × 10^6^	4.15 × 10^9^	1.49 × 10^16^
PVC [48]	g	1.38 × 10^5^	1.14 × 10^10^	1.57 × 10^15^
Wood [46]	g	6.00 × 10^6^	8.80 × 10^8^	5.28 × 10^15^
Manure (purchased) [45]	g	5.74 × 10^9^	9.70 × 10^7^	5.57 × 10^17^
			F	5.92 × 10^17^
			U	6.24 × 10^17^
Products				
Biogas	J	1.23 × 10^13^		
Ammonia nitrogen	g	7.69 × 10^6^		
Phosphate phosphorus	g	9.04 × 10^5^		
EYR	1.05 × 10^0^			
ELR	1.82 × 10			
ESI	5.80 × 10^−2^			

**Table 3 ijerph-18-07538-t003:** Emergy calculation of biogas subsystem (after).

	Unit	Units/Year	Transformity (sej/Unit)	Emergy (sej/Year)
Renewable resources (R)				
Sunlight	J	2.37 × 10^13^	1	2.37 × 10^13^
Water [44]	m^3^	1.01 × 10^5^	3.23 × 10^11^	3.26 × 10^16^
Electricity (from the system) [38]	J	4.68 × 10^10^	1.65 × 10^5^	7.72 × 10^15^
Manure (from the system) [45]	g	5.74 × 10^9^	9.70 × 10^7^	5.57 × 10^17^
			R	5.97 × 10^17^
Purchased resources (F)				
Electricity(purchased) [38]	J	0	1.65 × 10^5^	0
Plastic [46]	g	4.15 × 10^5^	5.87 × 10^9^	2.44 × 10^15^
Zinc [47]	g	2.14 × 10^5^	1.58 × 10^10^	3.37 × 10^15^
Steel [46]	g	3.58 × 10^6^	4.15 × 10^9^	1.49 × 10^16^
PVC [48]	g	1.38 × 10^5^	1.14 × 10^10^	1.57 × 10^15^
Wood [46]	g	6.00 × 10^6^	8.80 × 10^8^	5.28 × 10^15^
Manure (purchased) [45]	g	0	9.70 × 10^7^	0
			F	2.75 × 10^16^
			U	6.24 × 10^17^
Products				
Biogas	J	1.23 × 10^13^		
Ammonia nitrogen	g	7.68 × 10^6^		
Phosphate phosphorus	g	9.04 × 10^5^		
EYR	2.27 × 10			
ELR	4.61 × 10^−2^			
ESI	4.92 × 10^2^			

**Table 4 ijerph-18-07538-t004:** Emergy calculation of constructed wetland subsystem.

	Unit	Units/Year	Transformity (sej/Unit)	Emergy (sej/Year)
Renewable resources (R)				
Sunlight	J	1.57 × 10^17^	1	1.57 × 10^17^
Rain [50]	J	7.44 × 10^13^	1.42 × 10^4^	1.06 × 10^18^
Wastewater inflow [51]	J	2.97 × 10^16^	3.80 × 10^6^	1.13 × 10^23^
			R	1.13 × 10^23^
Non-renewable resources (N)				
Soil [52]	J	1.04 × 10^15^	7.40 × 10^4^	7.70 × 10^19^
Organic compost [52]	J	5.96 × 10^15^	7.40 × 10^4^	4.41 × 10^20^
Wood fiber peat [52]	J	7.79 × 10^15^	7.40 × 10^4^	5.76 × 10^20^
Activated sludge [52]	J	4.29 × 10^14^	7.40 × 10^4^	3.17 × 10^19^
			N	1.13 × 10^21^
Purchased resources (F)				
Gravel [38]	g	1.12 × 10^12^	1.00 × 10^9^	1.12 × 10^21^
Sand [38]	g	1.58 × 10^12^	1.00 × 10^9^	1.58 × 10^21^
Vegetation [53]	$	2.62 × 10^7^	1.16 × 10^13^	3.04 × 10^20^
Iron ore powder [54]	g	1.58 × 10^11^	1.28 × 10^9^	2.02 × 10^20^
PC liner [38]	J	7.85 × 10^13^	1.11 × 10^5^	8.71 × 10^18^
PE pipe [38]	J	1.73 × 10^13^	1.11 × 10^5^	1.92 × 10^18^
Geotextile [38]	J	1.71 × 10^13^	1.11 × 10^5^	1.90 × 10^18^
Steel griller [55]	g	4.94 × 10^7^	4.13 × 10^9^	2.04 × 10^17^
Bricks and cement [55]	g	1.39 × 10^11^	1.97 × 10^9^	2.74 × 10^20^
Machinery [52]	g	4.12 × 10^8^	6.70 × 10^9^	2.76 × 10^18^
Electricity [38]	J	1.23 × 10^15^	1.66 × 10^5^	1.96 × 10^20^
			F	3.70 × 10^21^
			U	1.18 × 10^23^
Products				
Treated wastewater	J	2.48 × 10^16^		
EYR	3.19 × 10			
ELR	4.27 × 10^−2^			
ESI	7.47 × 10^2^			

**Table 5 ijerph-18-07538-t005:** Emergy calculation of rainwater garden subsystem (before).

	Unit	Units/Year	Transformity (sej/Unit)	Emergy (sej/Year)
Renewable resources (R)				
Sun	J	1.05 × 10^18^	1	1.05 × 10^18^
Rain [50]	J	1.19 × 10^16^	1.42 × 10^4^	1.69 × 10^20^
Fertilizers (from the system) [38]	g	0	8.28 × 10^9^	0
Municipal water (from the system) [55]	J	0	4.31 × 10^5^	0
			R	1.69 × 10^20^
Non-renewable resources (N)				
Net topsoil loss [56]	J	3.49 × 10^13^	1.24 × 10^5^	4.33 × 10^18^
			N	4.33 × 10^18^
Purchased resources (F)				
Constructed debris disposal [54]	g	1.49 × 10^14^	1.94 × 10^8^	2.89 × 10^22^
Stone [57]	g	1.90 × 10^13^	1.64 × 10^9^	3.12 × 10^22^
Topsoil [57]	J	3.73 × 10^13^	1.68 × 10^9^	6.26 × 10^22^
Fertilizers(purchased) [38]	g	3.25 × 10^9^	8.28 × 10^9^	2.69 × 10^19^
Plant materials [58]	$	1.18 × 10^8^	1.16 × 10^13^	1.37 × 10^21^
Municipal water (purchased) [55]	J	9.22 × 10^10^	4.31 × 10^5^	3.97 × 10^16^
HDPE pipe [57]	g	3.45 × 10^9^	9.68 × 10^9^	3.34 × 10^19^
PVC drain [54]	g	9.04 × 10^9^	9.68 × 10^9^	8.75 × 10^19^
Steel drain filter [58]	g	1.12 × 10^9^	6.92 × 10^9^	7.74 × 10^18^
Hand tools [55]	g	2.13 × 10^9^	6.92 × 10^9^	1.47 × 10^19^
Heavy machinery [58]	$	4.30 × 10^8^	1.16 × 10^13^	4.98 × 10^21^
			F	1.29 × 10^23^
			U	1.29 × 10^23^
Products				
Water	J	4.88 × 10^15^		
EYR	1			
ELR	7.63 × 10^2^			
ESI	1.31 × 10^−3^			

**Table 6 ijerph-18-07538-t006:** Emergy calculation of rainwater garden subsystem (after).

	Unit	Units/Year	Transformity (sej/Unit)	Emergy (sej/Year)
Renewable resources (R)				
Sun	J	1.05 × 10^18^	1	1.05 × 10^18^
Rain [50]	J	1.19 × 10^16^	1.42 × 10^4^	1.69 × 10^20^
Fertilizers (from the system) [38]	g	3.25 × 10^9^	8.28 × 10^9^	2.69 × 10^19^
Municipal water (from the system) [55]	J	9.22 × 10^10^	4.31 × 10^5^	3.97 × 10^16^
			R	1.69 × 10^20^
Non-renewable resources (N)				
Net topsoil loss [56]	J	3.49 × 10^13^	1.24 × 10^5^	4.33 × 10^18^
			N	4.33 × 10^18^
Purchased resources (F)				
Constructed debris disposal [54]	g	1.49 × 10^14^	1.94 × 10^8^	2.89 × 10^22^
Stone [57]	g	1.90 × 10^13^	1.64 × 10^9^	3.12 × 10^22^
Topsoil [57]	g	3.73 × 10^13^	1.68 × 10^9^	6.27 × 10^22^
Fertilizers(purchased) [38]	g	0	8.28 × 10^9^	0
Plant materials [58]	$	1.18 × 10^8^	1.16 × 10^13^	1.37 × 10^21^
Municipal water (purchased) [55]	J	0	4.31 × 10^5^	0
HDPE pipe [57]	g	3.45 × 10^9^	9.68 × 10^9^	3.34 × 10^19^
PVC drain [54]	g	9.04 × 10^9^	9.68 × 10^9^	8.75 × 10^19^
Steel drain filter [58]	g	1.12 × 10^9^	6.92 × 10^9^	7.75 × 10^18^
Hand tools [55]	g	2.13 × 10^9^	6.92 × 10^9^	1.47 × 10^19^
Heavy machinery [58]	$	4.30 × 10^8^	1.16 × 10^13^	4.99 × 10^21^
			F	1.29 × 10^23^
			U	1.29 × 10^23^
Products				
Water	J	4.88 × 10^15^		
EYR	1			
ELR	7.62 × 10^2^			
ESI	1.31 × 10^−3^			

**Table 7 ijerph-18-07538-t007:** Emergy calculation of urban farm (before).

	Unit	Units/Year	Transformity (sej/Unit)	Emergy (sej/Year)
Renewable resources (R)				
Sunlight	J	1.70 × 10^16^	1	1.70 × 10^16^
Rain [50]	J	1.73 × 10^13^	1.42 × 10^4^	2.46 × 10^17^
Electricity (from the system) [38]	J	0	1.66 × 10^5^	0
Municipal water (from the system) [55]	J	0	4.31 × 10^5^	0
Compost (from the system) [59]	g	0	2.42 × 10^8^	0
			R	2.63 × 10^17^
Purchased resources (F)				
Mulch [46]	J	5.20 × 10^11^	2.76 × 10^4^	1.44 × 10^16^
Fuel [60]	g	1.19 × 10^7^	5.21 × 10^4^	6.21 × 10^11^
Organic matter [46]	J	8.46 × 10^11^	5.85 × 10^4^	4.95 × 10^16^
Straw [46]	J	3.25 × 10^12^	6.92 × 10^4^	2.25 × 10^17^
Cardboard [46]	J	6.26 × 10^11^	1.12 × 10^5^	7.01 × 10^16^
Seeds [37]	J	6.91 × 10^11^	2.88 × 10^5^	1.99 × 10^17^
Plastic [46]	g	8.24 × 10^6^	3.00 × 10^8^	2.47 × 10^15^
Wood [46]	g	6.46 × 10^7^	6.95 × 10^8^	4.49 × 10^16^
Brick [46]	g	3.83 × 10^5^	1.83 × 10^9^	7.01 × 10^14^
Aluminum [61]	g	1.09 × 10^7^	2.24 × 10^9^	2.45 × 10^16^
Steel [62]	g	2.77 × 10^4^	2.80 × 10^9^	7.76 × 10^13^
Rubber [46]	g	6.19 × 10^6^	3.40 × 10^9^	2.10 × 10^16^
Perlite [38]	kg	1.83 × 10^7^	3.56 × 10^9^	6.52 × 10^16^
Pesticide, neem oil [38]	kg	2.46 × 10^2^	1.26 × 10^12^	3.10 × 10^14^
Electricity (purchased) [38]	J	3.37 × 10^12^	1.66 × 10^5^	5.60 × 10^17^
Municipal water (purchased) [55]	J	6.05 × 10^10^	4.31 × 10^5^	2.61 × 10^16^
Compost (purchased) [59]	g	3.35 × 10^10^	2.42 × 10^8^	8.10 × 10^18^
			F	9.41 × 10^18^
			U	9.67 × 10^18^
Products				
Food	kg	9.54 × 10^6^		
EYR	1.03 × 10^0^			
ELR	3.58 × 10			
ESI	2.87 × 10^−2^			

**Table 8 ijerph-18-07538-t008:** Emergy calculation of urban farm (after).

	Unit	Units/Year	Transformity (sej/Unit)	Emergy (sej/Year)
Renewable resources (R)				
Sunlight	J	1.70 × 10^16^	1	1.70 × 10^16^
Rain [50]	J	1.73 × 10^13^	1.42 × 10^4^	2.46 × 10^17^
Electricity (from the system) [38]	J	3.37 × 10^12^	1.66 × 10^5^	5.59 × 10^17^
Municipal water (from the system) [55]	J	6.05 × 10^10^	4.31 × 10^5^	2.61 × 10^16^
Compost (from the system) [59]	g	3.35 × 10^10^	2.42 × 10^8^	8.11 × 10^18^
			R	8.95 × 10^18^
Purchased resources (F)				
Mulch [46]	J	5.20 × 10^11^	2.76 × 10^4^	1.44 × 10^16^
Fuel [60]	g	1.19 × 10^7^	5.21 × 10^4^	6.20 × 10^11^
Organic matter [46]	J	8.46 × 10^11^	5.85 × 10^4^	4.95 × 10^16^
Straw [46]	J	3.25 × 10^12^	6.92 × 10^4^	2.25 × 10^17^
Cardboard [46]	J	6.26 × 10^11^	1.12 × 10^5^	7.01 × 10^16^
Seeds [37]	J	6.91 × 10^11^	2.88 × 10^5^	1.99 × 10^17^
Plastic [46]	g	8.24 × 10^6^	3.00 × 10^8^	2.47 × 10^15^
Wood [46]	g	6.46 × 10^7^	6.95 × 10^8^	4.49 × 10^16^
Brick [46]	g	3.83 × 10^5^	1.83 × 10^9^	7.01 × 10^14^
Aluminum [61]	g	1.09 × 10^7^	2.24 × 10^9^	2.44 × 10^16^
Steel [62]	g	2.77 × 10^4^	2.80 × 10^9^	7.76 × 10^13^
Rubber [46]	g	6.19 × 10^6^	3.40 × 10^9^	2.10 × 10^16^
Perlite [38]	kg	1.83 × 10^7^	3.56 × 10^9^	6.51 × 10^16^
Pesticide, neem oil [38]	kg	2.46 × 10^2^	1.26 × 10^12^	3.10 × 10^14^
Electricity (purchased) [38]	J	0	1.66 × 10^5^	0
Municipal water (purchased) [55]	J	0	4.31 × 10^5^	0
Compost (purchased) [59]	g	0	2.42 × 10^8^	0
			F	7.17 × 10^17^
			U	9.67 × 10^18^
Products				
Food	kg	9.54 × 10^6^		
EYR	1.35 × 10			
ELR	8.01 × 10^−2^			
ESI	1.68 × 10^2^			

**Table 9 ijerph-18-07538-t009:** Emergy calculation of solar power subsystem (before).

	Unit	Units/Year	Transformity (sej/Unit)	Emergy (sej/Year)
Renewable resources (R)				
Sunlight	J	3.21 × 10^15^	1	3.21 × 10^15^
Rain [50]	J	7.60 × 10^7^	1.42 × 10^4^	1.08 × 10^12^
Water (from the system) [44]	g	0	1.59 × 10^6^	0
			R	3.21 × 10^15^
Purchased resources (F)				
Fuel [52]	g	1.64 × 10^6^	6.22 × 10^9^	1.02 × 10^16^
Oil [52]	J	1.74 × 10^8^	1.48 × 10^5^	2.58 × 10^13^
Coal [64]	J	2.98 × 10^9^	8.17 × 10^4^	2.44 × 10^14^
Natural gas [65]	J	4.81 × 10^8^	1.70 × 10^5^	8.18 × 10^13^
Uranium [66]	g	2.93 × 10^0^	1.68 × 10^11^	4.92 × 10^11^
Photoactive materials [65]	g	2.46 × 10^7^	5.76 × 10^11^	1.42 × 10^19^
Glass [65]	g	3.81 × 10^8^	8.00 × 10^9^	3.05 × 10^18^
Copper [66]	g	1.52 × 10^7^	1.02 × 10^11^	1.55 × 10^18^
Aluminum [66]	g	2.47 × 10^7^	5.73 × 10^9^	1.41 × 10^17^
Steel [66]	g	5.81 × 10^7^	1.24 × 10^10^	7.20 × 10^17^
EVA and plastics [64]	g	1.27 × 10^7^	6.22 × 10^9^	7.89 × 10^16^
Water (purchased) [44]	g	3.82 × 10^9^	1.59 × 10^6^	6.07 × 10^15^
			F	1.97 × 10^19^
			U	1.97 × 10^19^
Products				
Electricity	J	2.79 × 10^14^		
EYR	1.00 × 10^0^			
ELR	6.14 × 10^3^			
ESI	1.63 × 10^−4^			

**Table 10 ijerph-18-07538-t010:** Emergy calculation of solar power subsystem (after).

	Unit	Units/Year	Transformity (sej/Unit)	Emergy (sej/Year)
Renewable resources (R)				
Sunlight	J	3.21 × 10^15^	1	3.21 × 10^15^
Rain [50]	J	7.60 × 10^7^	1.42 × 10^4^	1.08 × 10^12^
Water (from the system) [44]	g	3.82 × 10^9^	1.59 × 10^6^	6.07 × 10^15^
			R	9.28 × 10^15^
Purchased resources (F)				
Fuel [52]	g	1.64 × 10^6^	6.22 × 10^9^	1.02 × 10^16^
Oil [52]	J	1.74 × 10^8^	1.48 × 10^5^	2.58 ×10^13^
Coal [64]	J	2.98 × 10^9^	8.17 × 10^4^	2.44 × 10^14^
Natural gas [65]	J	4.81 × 10^8^	1.70 × 10^5^	8.18 × 10^13^
Uranium [66]	g	2.93 × 10^0^	1.68 × 10^11^	4.92 × 10^11^
Photoactive materials [65]	g	2.46 × 10^7^	5.76 × 10^11^	1.42 × 10^19^
Glass [65]	g	3.81 × 10^8^	8.00 × 10^9^	3.05 × 10^18^
Copper [66]	g	1.52 × 10^7^	1.02 × 10^11^	1.55 × 10^18^
Aluminum [66]	g	2.47 × 10^7^	5.73 × 10^9^	1.41 × 10^17^
Steel [66]	g	5.81 × 10^7^	1.24 × 10^10^	7.20 × 10^17^
EVA and plastics [64]	g	1.27 × 10^7^	6.22 × 10^9^	7.89 × 10^16^
Water (purchased) [44]	g	0	1.59 × 10^6^	0
			F	1.97 × 10^19^
			U	1.97 × 10^19^
Products				
Electricity	J	2.79 × 10^14^		
EYR	1.00 × 10^0^			
ELR	2.12 × 10^3^			
ESI	4.71 × 10^−4^			

**Table 11 ijerph-18-07538-t011:** Emergy calculation of the complete Urban Self-circulation System.

	EYR	ELR	ESI
Biogas (before)	1.05 × 10^0^	1.82 × 10	5.80 × 10^−2^
Biogas (after)	2.27 × 10	4.61 × 10^−2^	4.92 × 10^2^
Constructed wetland	3.18 × 10	4.28 × 10^−2^	7.44 × 10^2^
Urban farm (before)	1.03 × 10^0^	3.58 × 10	2.87 × 10^−2^
Urban farm (after)	1.35 × 10	8.01 × 10^−2^	1.68 × 10^2^
Rainwater garden (before)	1	7.63 × 10^2^	1.31 × 10^−3^
Rainwater garden (after)	1	7.62 × 10^2^	1.31 × 10^−3^
Solar power subsystem (before)	1	6.14 × 10^3^	1.63 × 10^−4^
Solar power subsystem (after)	1	2.12 × 10^3^	4.71 × 10^−4^
Total system	1.86 × 10^0^	1.18 × 10^0^	1.57 × 10^0^
Core two subsystems	3.18 × 10	4.27 × 10^−2^	7.45 × 10^2^

## Data Availability

Data are contained within the article.

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
