# Peer review of "Multi-System Urban Waste-Energy Self-Circulation: Design of Urban Self-Circulation System Based on Emergy Analysis"

_ijerph, 2021, doi:10.3390/ijerph18147538_

Round 1

Reviewer 1 Report

This work addresses a very interesting and high-interest topic, it also provides a lot of information.

The abstract does not present any research results, it must be substantially improved.

In the introduction the self-circulation system is not adequately informed, it would be pertinent to link this with a general flow diagram of the proposal.
The methodology should be more explicit, such that the raw information that is available is linked and how it fits with the development of the proposal, in this aspect the presentation is very important.

In the development of the work, a clear link is not established about the mass and energy flows, so it is necessary to make flow diagrams and the respective mass and energy balances.

The conclusions are generic and are not qualitatively and quantitatively linked to the development of the research.This work addresses a very interesting and high-interest topic, it also provides a lot of information.

Author Response

Dear Editors and Reviewers:

Thank you for your letter and for the reviewer’s comments concerning our manuscript entitled “Multi-System Urban Waste-Energy Self-Circulation: Design of Urban Self-circulation System Based on Emergy Analysis”. (ijerph-1261169). Those comments are all valuable and very helpful for revising and improving my paper, as well as the important guiding significance to our researches.

I have studied comments carefully and have made correction which I hope meet with approval. Revised portion are marked using the “Track Changes” function in the paper. The main corrections in the paper and the responds to the reviewer’s comments are as flowing:

Responds to the reviewer’s comments:

  1. Response to comment: The abstract does not present any research results, it must be substantially improved. The conclusions are generic and are not qualitatively and quantitatively linked to the development of the research. This work addresses a very interesting and high-interest topic, it also provides a lot of information.

Response: I have re-analyzed the results of the study to draw a series of conclusions qualitatively and quantitatively. it is really true as you suggested that the study provides a wealth of information that the previous manuscript I did not dig into what this information represents meaning. Therefore I have rewritten the results, discussion and conclusion sections and briefly described them in the abstract.

  1. Response to comment: In the introduction the self-circulation system is not adequately informed, it would be pertinent to link this with a general flow diagram of the proposal.

Response: I have added a description of the self-circulation system in the introduction section and combined it with a general flow diagram of the proposal.

  1. Response to comment: The methodology should be more explicit, such that the raw information that is available is linked and how it fits with the development of the proposal, in this aspect the presentation is very important.

Response: The presentation of the methodology in the original manuscript was brief and incomplete. Therefore, I have completed the entire theoretical framework of the emergy evaluation and described the process and implications of the evaluation in the context of the development of the proposal and its raw information.

  1. Response to comment: In the development of the work, a clear link is not established about the mass and energy flows, so it is necessary to make flow diagrams and the respective mass and energy balances.

Response: I provide a more detailed description of the self-circulation system as a whole and the energy flow in each subsystem, including the methods taken by each subsystem and the input and output substances.

I tried my best to improve the manuscript and made these changes in the manuscript. These changes will not influence the content and framework of the paper.

I appreciate for Editors/Reviewers’ warm work earnestly, and hope that the correction will meet with approval.

Once again, thank you very much for your comments and suggestions.

Yours

Xiaoyu Xu

Reviewer 2 Report

This study proposes an urban self-circulation design using the updated emergy analysis method, the emergy output, the environmental load and sustainability as indicators. This analysis demonstrates that the system takes advantage of low total cost and environmental pollution, and high output.

However, the study does not provide any novelty, and the techniques utilized are not described or motivated thoroughly. For instance, the author does not clearly describe the biogas system used in this study and claims that nitrogen and phosphorus were generated from it, which has not been done so far because the nitrogen and phosphorus are not reduced during anaerobic digestion. Furthermore, most concepts used in this study are very generic and are not scientifically supported. Units are sometimes not assigned to their corresponding values, the figures are fuzzy, and the data collection process and its reliability were not presented. Additionally, no statistical analysis was done on the data. See below further comments regarding the manuscript.

Line 8: What city is referred to here, or alternatively the type of city (urban or rural), the size of the city, or other related characteristics.

I would recommend presenting key results or findings from this study in the abstract.

Figure 1 looks very fuzzy, I would suggest an increase of the font size and line width.
Lines 122 – 125: The anaerobic digestion, which I believe is the system to be used to turn organic waste into biogas, does not produce nitrogen or phosphorus. Nutrients are not reduced by the anaerobic digestion.

Line 126: what biogas subsystem is referred to here? Can you describe its operation, and provide its efficiency?

Lines 127 – 128: Biogas systems do not convert organic waste into nutrients and phosphorus.

Lines 140 – 142: What about the throughput of wetlands compared to the one of traditional wastewater treatment system, are wetland systems sustainable in big cities with high inputs of wastewater?

Line 168: What type of agricultural farming is suggested here, vertical or horizontal farming?

Line 182: Was a mass or energy balance performed in this system to back up this statement?

Lines 188 – 189: the author once again alludes the generation of nitrogen and phosphorus from a biogas system. Please explain how this system works and be more specific about the type of biogas system, as anaerobic digestion does not generate nitrogen or phosphorus.

Line 199: The local temperature in Rhodes Island varies between 48o to 85oF, which is not the ideal temperature range for anaerobic digestion for instance, which is an essential part of this process, how can this climate be suitable for it?

Table 1: why is the unit of measurement of water in J?

Line 285: Please remain consistent with the use of units. The unit of waste in Table 1 is Kg, and then g in line 285.

Lines 288-289: The energy output. The rates do not have a unit. The same in lines 291 and 293 for the environmental load rate and the loading rate, respectively.

Table 2: What does the unit “h” refer to?

Lines 300-301: What are the units of these values.

Line 305: Why is the unit of water expressed in J?

Although the emergy output rate unit is known, it is recommended to place the units next to the values wherever provided.

Figures 2, 3, and 4 are fuzzy, and should have error bars.

Author Response

Dear Editors and Reviewers:

Thank you for your letter and for the reviewer’s comments concerning our manuscript entitled “Multi-System Urban Waste-Energy Self-Circulation: Design of Urban Self-circulation System Based on Emergy Analysis”. (ijerph-1261169). Those comments are all valuable and very helpful for revising and improving my paper, as well as the important guiding significance to our researches.

I have studied comments carefully and have made correction which I hope meet with approval. Revised portion are marked using the “Track Changes” function in the paper. The main corrections in the paper and the responds to the reviewer’s comments are as flowing:

Responds to the reviewer’s comments:

  1. Response to comment: What city is referred to here, or alternatively the type of city (urban or rural), the size of the city, or other related characteristics.

Response: I added in the abstract that the system is suitable for small-scale cities as a complement to large energy production systems, and have duly emphasized this in the text.

  1. Response to comment: I would recommend presenting key results or findings from this study in the abstract.

Response: I have re-analyzed the results of the study to draw a series of conclusions qualitatively and quantitatively. it is really true as you suggested that the study provides a wealth of information that the previous manuscript I did not dig into what this information represents meaning. Therefore I have rewritten the results, discussion and conclusion sections and briefly described them in the abstract.

  1. Response to comment: Figure 1 looks very fuzzy, I would suggest an increase of the font size and line width.

Response: I redrew this diagram as a vector drawing instead of the original drawing.

  1. Response to comment: The anaerobic digestion, which I believe is the system to be used to turn organic waste into biogas, does not produce nitrogen or phosphorus. Nutrients are not reduced by the anaerobic digestion.

&Biogas systems do not convert organic waste into nutrients and phosphorus.

&The author once again alludes the generation of nitrogen and phosphorus from a biogas system. Please explain how this system works and be more specific about the type of biogas system, as anaerobic digestion does not generate nitrogen or phosphorus.

Response: I re-reviewed studies involving biogas systems, including "Emergy analysis of biogas production and electricity generation from small-scale agricultural digesters" and "The microbiology of anaerobic digesters", which provides an overview of anaerobic digesters. According to these articles, the products would include ammonia nitrogen and phosphate phosphorus, so I think this is plausible.

  1. Response to comment: what biogas subsystem is referred to here? Can you describe its operation, and provide its efficiency?

Response: In the article, I added information about the operation principle, input and output substances, and efficiency of the five subsystems including the biogas system, which is small-scale anaerobic digesters.

  1. Response to comment:  What about the throughput of wetlands compared to the one of traditional wastewater treatment system, are wetland systems sustainable in big cities with high inputs of wastewater?

Response: I added information about the operation principle, input and output substances, and efficiency of the five subsystems including the constructed wetland system, which is small-scale shallow ponds or channels planted with aquatic plants.

  1. Response to comment: What type of agricultural farming is suggested here, vertical or horizontal farming?

Response: I added information about the operation principle, input and output substances, and efficiency of the five subsystems including the urban farm system, which is horizontal farming.

  1. Response to comment:  Was a mass or energy balance performed in this system to back up this statement?

Response: Yes. I performed energy balance in this system. I have added some words to mention this point.

  1. Response to comment:  The local temperature in Rhodes Island varies between 48o to 85oF, which is not the ideal temperature range for anaerobic digestion for instance, which is an essential part of this process, how can this climate be suitable for it?

Response: I checked the energy systems in Providence and Rhodes Island and found that there are several (about 5 around Providence) biogas systems that have been working well for several years. Therefore, biogas systems can still be effective in the New England climate.

  1. Response to comment: Table 1: why is the unit of measurement of water in J? 

& Why is the unit of water expressed in J?

Response: This is mainly due to the fact that the transformity of individual substances is derived from the provisions of previous studies in which the transformity of water was set in sej/J.

  1. Response to comment: Please remain consistent with the use of units. The unit of waste in Table 1 is Kg, and then g in line 285.

Response: Thank you for the suggestion. I have changed the unit into Kg in line 285.

  1. Response to comment: The energy output. The rates do not have a unit. The same in lines 291 and 293 for the environmental load rate and the loading rate, respectively.

& Although the emergy output rate unit is known, it is recommended to place the units next to the values wherever provided.

&What are the units of these values.

Response: I reviewed a series of relevant studies on emergy assessment and found that none of the three metrics used as ratios were labeled with units. So I think perhaps they should be unlabeled.

  1. Response to comment: What does the unit “h” refer to?

Response: I'm sorry, there is a clerical error here. It should be J.

  1. Response to comment: Figures 2, 3, and 4 are fuzzy, and should have error bars.

Response: The original three charts are not complete in their presentation and analysis of information. Therefore, I redid the three graphs and rewrote the three sections involving the analysis of their information: results, discussion, and conclusion.

I tried my best to improve the manuscript and made these changes in the manuscript. These changes will not influence the content and framework of the paper.

I appreciate for Editors/Reviewers’ warm work earnestly, and hope that the correction will meet with approval.

Once again, thank you very much for your comments and suggestions.

Yours

Xiaoyu Xu

Round 2

Reviewer 1 Report

Abstract must include the quantitative results obtained.
Conclusions should be reinforced with quantitative results.
With the above, more relevance is given to the indicators used in the methodology.

"The study used constructed wetland systems as a water purification system. Constructed 366 wetlands are artificial wastewater treatment systems consisting of shallow ponds or chan- 367 nels planted with aquatic plants, which rely on natural microbial, biological, physical and 368 chemical processes to treat wastewater [27] "Lines 365-369
Replace purification by depuration, they are wastewater.

The application of constructed wetlands must at least mention the use of surface, which constitutes a restriction generally associated with the availability of area by the city.
In addition, the availability of the area implies addressing the location of the wetlands in the city as a relevant issue.

"For waste utilization improvement, the urban self-recycling system in this paper contains two waste treatment subsystems: a biogas system (organic waste to biogas) and a constructed wetland system (wastewater to municipal available water)". Lines 111-113

It is necessary to specify the reuse of wastewater as a source of water, in what ways, irrigation source of drinking water, filling of aquifers, etc.

Legend to figure 2.

Figure 2 should be complemented with two more figures, one that explains in more detail the flows of water and derivatives and the other of the biogas. Keep figure 2.

"The downtown area of ​​the city of Providence is a standard and complete area, repre- sentative of small to medium-sized towns: 10,000 people live, spend money, and work within it. The downtown area has a robust mix of commercial, office , residential, and ed- ucational functions where people can live a regular city life. " (lines 327-329)
This population is not consistent with the Annual Water Volume, 1.01E + 05 m3, this water consumption corresponds to approximately 2500 people. It is necessary to vicular data of water and the population or clarify the point.

Table 2 Emergy calculation of biogas subsystem (before), to better explain this table, I recommend including Explanatory bullet.
The other alternative is to divide this table, it is a lot of data at one time.

Despite insisting on a major correction, the work presents great advances and I believe that with a second correction it would be in a position to be published.

Author Response

Dear Reviewer: Thank you for your letter and for the reviewer’s comments concerning our manuscript entitled “Multi-System Urban Waste-Energy Self-Circulation: Design of Urban Self-circulation System Based on Emergy Analysis”. (ijerph-1261169). Those comments are all valuable and very helpful for revising and improving my paper, as well as the important guiding significance to our researches. I have studied comments carefully and have made correction which I hope meet with approval. Revised portion are marked using the “Track Changes” function in the paper.

The main corrections in the paper and the responds to the reviewer’s comments are as flowing:

Responds to the reviewer’s comments:

  1. Response to comment: Abstract must include the quantitative results obtained. Conclusions should be reinforced with quantitative results. With the above, more relevance is given to the indicators used in the methodology.
    Response: I made changes to the conclusion section, by adding quantitative results, and made corresponding changes to the summary section.
  2.  Response to comment: Replace purification by depuration, they are wastewater.
    Response: Yes, it is more accurate to use this word depuration and I have modified it in the article. Thanks for the suggestion.
  3. Response to comment: The application of constructed wetlands must at least mention the use of surface, which constitutes a restriction generally associated with the availability of area by the city. In addition, the availability of the area implies addressing the location of the wetlands in the city as a relevant issue. & It is necessary to specify the reuse of wastewater as a source of water, in what ways, irrigation source of drinking water, filling of aquifers, etc. & Table 2 Emergy calculation of biogas subsystem (before), to better explain this table, I recommend including Explanatory bullet. The other alternative is to divide this table, it is a lot of data at one time.
    Response: These comments made me realize that the article was not clear enough in describing the operation of the various subsystems. Therefore, I have added system diagrams for each subsystem and explained them in text.
  4. Response to comment: Legend to figure 2. Figure 2 should be complemented with two more figures, one that explains in more detail the flows of water and derivatives and the other of the biogas. Keep figure 2.
    Response: I have added the legend and two more figures to explain the whole system in more details.
  5. Response to comment: This population is not consistent with the Annual Water Volume, 1.01E + 05 m3, this water consumption corresponds to approximately 2500 people. It is necessary to vicular data of water and the population or clarify the point.
    Response: There is a misunderstanding here. 1.01E+05 is the amount of water entered into the biogas system, not the amount of water used in Downtown Providence.
    I tried my best to improve the manuscript and made these changes in the manuscript. These changes will not influence the content and framework of the paper. I appreciate for your warm work earnestly, and hope that the correction will meet with approval. Once again, thank you very much for your comments and suggestions this time and last time. They are helpful and made my article great advances.
    Yours
    Xiaoyu Xu

Reviewer 2 Report

Thank you for improving the quality of the manuscript as per the comments, it got a lot better. However, for future works, I would recommend a critical review and a thorough description of each process involved. 

Author Response

Dear Reviewer:

Thank you for your letter and for the reviewer’s comments concerning our manuscript entitled “Multi-System Urban Waste-Energy Self-Circulation: Design of Urban Self-circulation System Based on Emergy Analysis”. (ijerph-1261169). Those comments are all valuable and very helpful for revising and improving my paper, as well as the important guiding significance to our researches.

I have studied comments carefully and have made correction which I hope meet with approval. Revised portion are marked using the “Track Changes” function in the paper. The main corrections in the paper and the responds to the reviewer’s comments are as flowing:

Responds to the reviewer’s comments:

Thank you for improving the quality of the manuscript as per the comments, it got a lot better. However, for future works, I would recommend a critical review and a thorough description of each process involved.

Response: Thank you for your advice. Regarding this revision, I realized that the article was not clear enough in describing each subsystem as well as the total system, and the textual descriptions were not well integrated with the content of the tables. Therefore, I added 7 figures (two to complement the total system, one for each subsystem). The five subsystem diagrams are system diagrams about the role of each input energy resources in the system, and are supplemented with textual descriptions, which hopefully will make the descriptions clearer. In addition, I have also re-reviewed the article and revised some inaccurate parts, including a major change to the conclusion section.

I tried my best to improve the manuscript and made these changes in the manuscript. These changes will not influence the content and framework of the paper. I appreciate for your warm work earnestly, and hope that the correction will meet with approval.

Once again, thank you very much for your comments and suggestions this time and last time. They are helpful and made my article great advances.

Yours

Xiaoyu Xu

This manuscript is a resubmission of an earlier submission. The following is a list of the peer review reports and author responses from that submission.